# Dendritic Cell-Based Therapeutic Immunization Induces Th1/Th17 Responses and Reduces Fungal Burden in Experimental Sporotrichosis

**DOI:** 10.3390/microorganisms13102351

**Published:** 2025-10-14

**Authors:** Juliana Aparecida Jellmayer, Adriana Fernandes de Deus, Matheus Ricardo Curti Gonçalves, Lucas Souza Ferreira, Francine Alessandra Manente, Larissa Silva Pinho Caetano, Fernanda Luiza Piccineli, Thais Zamberço dos Reis Genari, Beatriz da Cunha Saçaki, Tarcila Pavicic Catalan de Oliveira Campos, Deivys Leandro Portuondo, Alexander Batista-Duharte, Iracilda Zeppone Carlos

**Affiliations:** 1Department of Clinical Analysis—School of Pharmaceutical Sciences, São Paulo State University—UNESP, Araraquara 14800-901, SP, Brazil; jujellmayer@yahoo.com.br (J.A.J.); adriana.deus@unesp.br (A.F.d.D.); mcurtivet@gmail.com (M.R.C.G.); gigabreath@hotmail.com (L.S.F.); famanente@gmail.com (F.A.M.); larissascaetano@gmail.com (L.S.P.C.); fernanda.piccineli@unesp.br (F.L.P.); bc.sacaki@unesp.br (B.d.C.S.); tarcila.catalan@unesp.br (T.P.C.d.O.C.); deivysleandro@gmail.com (D.L.P.); 2Department of Cell Biology—Physiology and Immunology, University of Cordoba, 14004 Córdoba, Spain; 3Immunology and Allergy Group (GC01)—Maimonides Biomedical Research Institute of Cordoba (IMIBIC), Reina Sofia University Hospital, 14004 Córdoba, Spain

**Keywords:** *Sporothrix schenckii*, vaccine, dendritic cells, bone-marrow-derived dendritic cells, sporotrichosis

## Abstract

Sporotrichosis is a globally distributed mycosis caused by thermally dimorphic fungi of the *Sporothrix schenckii* species complex. In Brazil, sporotrichosis is considered endemic and is usually acquired through zoonotic transmission from infected cats. The clinical manifestations may be cutaneous, lymphocutaneous, or systemic, the latter being more commonly observed in immunosuppressed patients. The limited effectiveness of antifungal treatments against this mycosis, particularly in immunocompromised individuals, has led to the search for more effective and safer therapies. Based on several studies demonstrating the efficient use of dendritic cells as tools for the development of antifungal vaccines, this work aimed to evaluate the protective capacity of bone marrow-derived dendritic cells (BMDCs) activated with cell wall proteins of *S. schenckii* (SsCWP) in mice infected with *S. schenckii sensu stricto*. BMDCs were stimulated with SsCWP and analyzed for the surface expression of costimulatory molecules as well as proinflammatory cytokine secretion. Subsequently, mice were vaccinated once or twice to assess immunogenicity, and finally, the therapeutic effect of BMDCs on *S. schenckii* infection was evaluated. Our results show that SsCWP was able to activate BMDCs. Immunization of healthy mice with SsCWP-stimulated BMDCs induced a balanced Th1/Th17-based immune response. Vaccination of mice previously infected with *S. schenckii* induced a mixed Th1/Th17 response and reduced fungal burden in the spleen. Overall, these findings demonstrate that therapeutic vaccination with SsCWP-stimulated BMDCs improves fungal control, supporting the notion that dendritic cells represent a promising therapeutic strategy against sporotrichosis.

## 1. Introduction

Fungal infections are an important cause of human morbidity and mortality and represent a growing concern due to the increasing use of broad-spectrum antifungal and immunosuppressive therapies [1]. Current treatments are often limited by prolonged treatment durations and the high toxicity of the available drugs, which can lead to significant side effects [2]. Consequently, the past decade has seen many studies focused on developing vaccines against endemic and opportunistic fungi [3,4,5,6,7]. Sporotrichosis is a subacute or chronic infection caused by thermo-dimorphic fungi of the *Sporothrix* genus. Although the disease has a cosmopolitan distribution, it predominantly occurs in tropical and subtropical regions and is considered the most frequent subcutaneous mycosis in Latin America, where it is endemic [8,9]. Classically, infection follows the traumatic inoculation of contaminated soil, plants, or organic material into the skin or mucosal tissues. Alternatively, zoonotic transmission may occur through scratches or bites from infected cats [10].

To minimize the damage caused by fungal infections, the human body displays a complex array of defense mechanisms, in which innate immunity plays a crucial role. Dendritic cells (DCs) serve as an early defense mechanism in various organs [11] and provide an essential bridge between innate and adaptive immunity. They are central players in the immune system, operating at the interface of innate and adaptive immunity. In both mice and humans. DCs are involved in the surveillance of diverse pathogens and in responding to microenvironmental tissue damage, and they possess specialized features that enable them to capture efficiently, process, and present antigens, in addition to their unique role in the activation, polarization, and regulation of adaptive immune responses [12,13,14].

Previously, our laboratory demonstrated the presence of a Th1 and Th17-type cellular immune response throughout the entire course of experimental sporotrichosis infection, as well as the participation of a Th2 and Tregs response in the advanced stages of the disease [15,16,17]. Subsequently, in other studies, we demonstrated that DCs stimulated with *S. schenckii* yeasts or their exoantigen were able to induce a mixed Th1/Th17 response in vitro [18]. We, along with other groups, have demonstrated that many proteins located on the cell wall of *S. schenckii* are important inducers of both antibody and cell-mediated immune responses, making them promising candidates for prophylactic and therapeutic strategies against sporotrichosis [5,19,20,21,22].

Depending on the DC subset and the type of stimulus received, DCs orchestrate the nature of subsequent T-cell responses. Driven by discrete sets of transcription factors and dependent on the DC growth factor Flt3-ligand, DC subsets develop from committed DC precursors (CDPs) in the bone marrow (BM). These precursors give rise to conventional DCs (cDCs), the subset with the most potent antigen-presenting capacity [23,24]. Upon activation, antigen-loaded cDCs initiate the differentiation of antigen-specific T cells into effector T cells with distinct functions and cytokine profiles. DC maturation is associated with a wide variety of cellular changes, including reduced antigen capture activity, increased expression of MHC class II and costimulatory molecules, the acquisition of chemokine receptors that guide migration, and the secretion of cytokines that control T-cell differentiation [25,26,27].

Whole cells and exoantigens have been tested in vitro, in prophylactic models, and in immunogenicity or even therapeutic studies; however, the complete set of cell wall proteins has never been evaluated as a proof of concept for their immunotherapeutic role [18,28,29]. Accordingly, in this study, we evaluated, for the first time, the therapeutic potential of SsCWP-stimulated BMDCs, comparing them with unstimulated BMDCs in an experimental mouse model of sporotrichosis.

## 2. Materials and Methods

Male BALB/c mice (5–6 weeks old) were purchased from the Multidisciplinary Center for Biological Research (CEMIB), University of Campinas, São Paulo, Brazil. The animals were housed in individually ventilated cages under controlled ambient temperature and a 12:12 h light/dark cycle. Food and water were provided ad libitum. All animal procedures followed the guidelines of the Brazilian College of Animal Experimentation (COBEA) and were approved by the Research Ethics Committee of the School of Pharmaceutical Sciences, São Paulo State University (UNESP), Araraquara CEUA/FCF/CAr. 07/2016), on 18 August 2016.

### 2.1. Microorganism and Culture Conditions

*S. schenckii sensu stricto* ATCC 16345, originally isolated from a human case of diffuse pulmonary infection (Baltimore, MD, USA) and provided by the Oswaldo Cruz Foundation (Rio de Janeiro, Brazil), was used in all experiments. For infection, fungal mycelium grown on Mycosel agar was transferred to brain-heart infusion (BHI) broth (Difco Laboratories, Detroit, MI, USA) and cultured for 6 days at 37 °C with constant shaking (150 rpm). An inoculum of 10^7^ yeast cells was then transferred to fresh medium and maintained under the same conditions for 5 additional days to ensure maximal mycelium-to-yeast conversion during logarithmic growth.

### 2.2. Extraction of the SsCWP

Extraction of SsCWPs was performed as previously described by Portuondo et al. [5], with minor modifications. The resulting pellets were washed with ice-cold acetone, dried in a SpeedVac^®^, and reconstituted in phosphate-buffered saline (PBS, pH 7.2–7.4). Protein concentration was determined using the BCA assay (Pierce).

### 2.3. Preparation of the Heat-Killed (HKss)

Heat-killed *S. schenckii* (HKss) cells were prepared from the same 5-day-old culture of the fungus in the brain-heart infusion broth used for animal infection in each respective experiment. Yeast cells were separated from the supernatant by centrifugation at 200× *g* for 5 min at room temperature, washed twice with 8 mL of sterile phosphate-buffered saline (PBS, pH 7.4), resuspended, and adjusted to 2.5 × 10^8^ yeast cells/mL in PBS. The suspension was incubated for 1 h in a 60 °C water bath and then stored at 2–8 °C until use. A working suspension was obtained by diluting the stock suspension 1:10 in Roswell Park Memorial Institute (RPMI) complete medium (RPMI-1640 supplemented with 20 μM 2-β-mercaptoethanol, 100 U/mL penicillin and streptomycin, 2 mM L-glutamine, and 5% fetal calf serum). To verify the efficiency of the heat-killing process, 100 μL aliquots from each preparation were plated on Mycosel agar and checked for colony-forming unit (CFU) growth before use.

### 2.4. BMDCs Generation

After euthanasia, bone marrow precursor cells were extracted from the femurs and tibias of BALB/c mice and resuspended in RPMI-1640 medium (Sigma, Waltham, MA, USA) supplemented with 10% heat-inactivated fetal calf serum, 100 U/mL penicillin, 100 µg/mL streptomycin, 5 mM 2-mercaptoethanol, and 1 mM sodium pyruvate (R-10 medium). The protocol was made by Quinello et al. [29]. The BMDCs were analyzed by flow cytometry for the expression of myeloid DC markers CD11c and MHC-II (Appendix A).

### 2.5. Stimulation of BMDCs with SsCWPs

For stimulation assays, 1 × 10^7^ BMDCs were resuspended in 10 mL of R-10 medium and transferred to new culture flasks, where they were either left unstimulated or stimulated with 25, 50, or 100 µg/mL SsCWPs for 24 h. Following incubation, cells were analyzed by flow cytometry for the expression of CD83 and the costimulatory molecules CD86, CD80, and CD40. Culture supernatants were collected and stored at −80 °C for subsequent cytokine quantification.

### 2.6. Flow Cytometry

BMDCs were washed with PBS containing 1% bovine serum albumin (BSA; Sigma) and then 1 × 10^6^ cells were stained with the following anti-mouse monoclonal antibodies (mAbs; BD Biosciences, Franklin Lakes, NJ, USA): PE anti-CD83 (clone Michel-19), PE-Cy7 anti-CD86 (clone GL1), PE anti-CD80 (clone 16-10A1), PE an-ti-CD40 (clone 3/23), FITC anti-CD11c (clone HL3) and APC anti-I-A^b^ (MHC-II) (clone AF6-120.1). BMDCs were identified by co-expression of CD11c and MHC-II, with unstained cells used as controls for gate positioning. Activation markers were evaluated within the CD11c^+^MHC-II^+^ population under unstimulated and SsCWP-stimulated conditions, and results were expressed as median fluorescence intensity (MFI). Analysis was performed using BD CSampler Software version 227.4.

### 2.7. Cytokine Detection in Culture Supernatant

Cytokines released into the supernatant of these cell cultures—TNF-α, IL-1β, IL-6, and IL-12—were quantified by enzyme-linked immunosorbent assay (ELISA) using 96-well plates, following the manufacturer’s instructions (DuoSet ELISA Kit, R&D Systems, Minneapolis, MN, USA). Absorbance was measured at 450 nm using a Multiskan Ascent microplate reader (Thermo Labsystems, Helsinki, Finland).

### 2.8. Vaccination of BALB/c Mice with BMDCs or SsCWP-Stimulated BMDCs

Male BALB/c mice (6–8 weeks old) were randomly assigned into five groups: (i) PBS control, (ii) unstimulated BMDCs administered subcutaneously in the upper dorsal region on day 0 (single dose), (iii) unstimulated BMDCs administered on days 0 and 7 (two doses), (iv) SsCWP-stimulated BMDCs administered on day 0 (single dose), and (v) SsCWP-stimulated BMDCs administered on days 0 and 7 (two doses). Before injection, BMDCs were stimulated with SsCWP (50 µg/mL, 24 h). The selection of 50 µg/mL SsCWP was based on the preliminary dose–response experiments (25, 50, 100 µg/mL) showing that 50 µg/mL induced robust DC activation and cytokine secretion, with no further significant advantage at 100 µg/mL. Fourteen days after the initial immunization, mice were euthanized, and spleens were collected for the assessment of Th1/Th17 immune responses.

### 2.9. Total Splenocytes

Spleens of treated mice were aseptically removed and passed through a 100-µm cell strainer into a Petri dish containing 2 mL of RPMI-1640 medium supplemented with 20 µM 2-β-mercaptoethanol, 100 U/mL penicillin and streptomycin, 2 mM L-glutamine, and 5% fetal calf serum (hereafter referred to as RPMI), using the plunger of a syringe. For red blood cell lysis, 6 mL of 0.17 M ammonium chloride solution was added to the resulting suspension, which was then incubated on ice for 5 min. Splenocytes were separated by centrifugation at 300× *g* for 5 min at 4 °C, washed once with 3 mL of RPMI, and resuspended in 1 mL of the same medium. Cell concentration and viability were determined by Trypan Blue exclusion using a hemocytometer under a light microscope.

### 2.10. Th1 and Th17 Analysis by Flow Cytometry

Viable splenocytes or cells from lymph nodes were adjusted to 1 × 10^7^ cells/mL in RPMI-1640 (Sigma-Aldrich, Taufkirchen, Germany) supplemented with 2 mM L-glutamine, 100 U/mL penicillin, 100 µg/mL streptomycin, and 10% fetal calf serum (RPMI complete). The following anti-mouse monoclonal antibodies (mAbs) were used: anti-CD16/CD32 (clone 93), FITC anti-CD3 (clone 17A2), APC anti-CD4 (clone RM4-5), PE anti-IL-17 (clone eBio17B7), PE-Cy7 anti-IFN-γ (clone XMG1.2) and the respective isotype controls. The cells were assessed for the frequency of Th1 (IFN-γ^+^), Th17 (IL-17^+^), and Th1/Th17 (IFN-γ^+^IL-17^+^) populations. Briefly, the cells were first stained for extracellular markers, then fixed and permeabilized using the eBioscience Intracellular Fixation & Permeabilization Buffer Set, followed by staining for the transcription factor Foxp3. Intracellular IFN-γ and IL-17 were detected in cultures of splenocytes or lymph node cells after in vitro stimulation with a cell stimulation cocktail containing phorbol 12-myristate 13-acetate (PMA) and ionomycin to induce cytokine production, along with Brefeldin A and Monensin to allow intracellular retention of cytokines (eBioscience). Events were acquired on a BD Accuri C6 flow cytometer (BD Biosciences) and analyzed using the manufacturer’s proprietary software.

### 2.11. Th1/Th2/Th17-Related Cytokines Analysis by Cytometric Bead Array (CBA)

Cytokine levels in splenocyte culture supernatants, obtained from a separate culture stimulated with heat-killed *S. schenckii* (HKss) without Monensin/Brefeldin A or left unstimulated, were quantified using the BD CBA Mouse Th1/Th2/Th17 Cytokine Kit (BD Biosciences, San Jose, CA, USA). This assay enables simultaneous detection of IL-2, IL-4, IL-6, IFN-γ, TNF, IL-17A, and IL-10 in a single sample. Procedures followed the manufacturer’s protocol, and data were acquired on a BD Accuri C6 flow cytometer (BD Biosciences).

### 2.12. Infection Model

A yeast suspension was prepared in PBS, and each mouse was inoculated subcutaneously in the left hind footpad with 1.6 × 10^7^ yeast cells in 0.02 mL of sterile PBS. At 72 h post-infection, mice were vaccinated with either BMDCs, SsCWP-stimulated BMDCs, or PBS as the control group. Fourteen days after infection, the animals were euthanized to evaluate the potential therapeutic effect by assessing fungal burden in the spleen and local lymph node, as well as the stimulation of the immune response, as described.

### 2.13. Fungal Load in the Spleen and Popliteal Lymph Node

The fungal load in the spleen and popliteal lymph nodes was determined by counting CFUs recovered from macerated organ homogenates. Organ homogenates were serially diluted, plated on Mycosel^TM^ agar, and examined after 7 days to determine CFU counts.

### 2.14. Statistical Analysis

Statistical analyses were performed using GraphPad Prism version 6.01. Student’s *t*-test or one- or two-way analysis of variance (ANOVA), followed by Tukey’s or Sidak’s multiple comparisons test, was applied when appropriate. Differences were considered statistically significant at *p* ≤ 0.05. Data are presented as mean ± SD. Each experiment was conducted with 4–10 mice (typically five); the exact number used in each experiment is provided in the corresponding figure legend.

## 3. Results

### 3.1. SsCWPs Induce BMDC Activation

BMDCs were generated by culturing bone marrow precursors in GM-CSF-supplemented RPMI medium, with ~80% successfully differentiating into BMDCs as confirmed by CD11c and MHC-II expression. Stimulation with 25–100 µg/mL SsCWPs increased the expression of the costimulatory markers CD83, CD80, CD86, and CD40 compared with unstimulated controls (Figure 1A–D). Stimulation with SsCWP additionally enhanced the secretion of proinflammatory cytokines TNF-α, IL-1β, IL-6, and IL-12 into the culture supernatant (Figure 2A–D), particularly at 50 and 100 µg/mL, indicating effective BMDC activation. Except for IL-12, cytokine secretion was significantly higher at 50 and 100 µg/mL compared with 25 µg/mL. For all subsequent experiments, BMDCs were stimulated with 50 µg/mL SsCWPs.

### 3.2. Single BMDC Immunization, With or Without SsCWP Stimulation, Induces a Balanced Th1/Th17 Response

The gating strategy used to identify Th17 and Th1 cell subsets is shown in Figure 3A. CD3^+^CD4^+^ T lymphocytes were first gated, and the frequencies of IFN-γ^+^ (Th1) and IL-17A^+^ (Th17) populations were subsequently determined. As shown in Figure 3B, immunization with BMDCs significantly increased the frequency of IL-17A^+^ cells compared with the control group. Similarly, Figure 3C shows that IFN-γ^+^ cell frequencies were also elevated in mice immunized with either unstimulated or SsCWP-stimulated BMDCs, with the strongest responses observed after a single immunization. The analysis of double-positive IL-17A^+^/IFN-γ^+^ cells (Figure 3D) revealed that all immunized groups displayed higher frequencies compared to the control, with the single-dose unstimulated BMDC group exhibiting the highest proportion of these cells. Together, these results confirm that immunization with BMDCs, whether unstimulated or stimulated with SsCWP, promotes the differentiation of both Th1 and Th17 subsets, while an additional dose does not further enhance this response.

Interestingly, this pattern differed from the systemic response, as cytokine quantification in splenocyte culture supernatants after 48 h of stimulation with heat-inactivated *S. schenckii* revealed distinct profiles depending on the vaccine used. Both groups immunized with DCs (unstimulated and SsCWP-stimulated) showed significantly higher levels of IL-10 compared with controls (Figure 4A), *p* < 0.05). IL-17 production (Figure 4B) was significantly increased in both groups compared with controls (*p* < 0.05), with a stronger effect observed in the DC/SsCWP group (*p* < 0.01). No significant differences were found in TNF-α levels among groups (Figure 4C). IFN-γ secretion (Figure 4D) was significantly elevated in both vaccinated groups relative to controls (*p* < 0.05). IL-6 levels (Figure 4E) were significantly higher in the DC group compared with controls (*p* < 0.01), while IL-4 production (Figure 4F) did not differ among groups. Finally, IL-2 secretion (Figure 4G) was significantly increased only in the DC group (*p* < 0.05).

### 3.3. BMDCs, Whether Unstimulated or SsCWP-Stimulated, Promote Th17/Th1 Responses and Decrease Fungal Burden in S. schenckii–Infected Mice

To evaluate the local immune response, lymph node cells from *S. schenckii*-infected mice were analyzed by flow cytometry using the same gating strategy (Figure 5A). The frequency of IL-17A^+^ cells (Figure 5B) was significantly reduced in mice immunized with unstimulated DCs compared with controls (*p* < 0.001), whereas vaccination with SsCWP-stimulated DCs did not differ from the control group. In contrast, IFN-γ^+^ cell frequencies (Figure 5C) were significantly increased in the DC/SsCWP group compared with both the control and DC groups (*p* < 0.01, *p* < 0.001). No significant differences were observed among groups in the IL-17A^+^IFN-γ^+^ double-positive population (Figure 5D). Overall, these results indicate that SsCWP-stimulated DCs enhance Th1 responses in draining lymph nodes and may improve protective immunity against *S. schenckii.*

The fungal burden in *S. schenckii*-infected mice was assessed by CFU counts in the popliteal lymph nodes and spleen following vaccination with unstimulated or SsCWP-stimulated DCs. In the lymph nodes (Figure 6A), no significant differences were detected among groups, although a trend toward reduced CFU counts was observed in vaccinated mice. In contrast, both DC/- and DC/SsCWP-vaccinated mice promoted a significant reduction in splenic fungal load compared with controls (Figure 6B), *p* < 0.05). These findings suggest that vaccination with DCs, irrespective of SsCWP stimulation, contributes to limiting systemic fungal dissemination, as evidenced by reduced spleen colonization, while also showing a non-significant tendency toward reduced fungal burden in lymph nodes.

## 4. Discussion

DC-based vaccines have emerged as a promising therapeutic approach due to their capacity to induce antigen-specific immune responses. Clinical and experimental studies have demonstrated that antigen-loaded DCs can effectively promote protective immunity against infections and cancer, as well as modulate detrimental responses in autoimmunity and transplantation [25,26,27,29,30]. Given their pivotal role in orchestrating protective immune responses during sporotrichosis [18,28,29,31,32], DCs represent an attractive platform for vaccine development. DC-targeted adjuvants are markedly more effective than conventional adjuvants in presenting antigens and stimulating antigen-specific immune responses, particularly T cell-mediated responses, owing to their highly specialized antigen presentation and immunoregulatory functions. For this reason, DCs have often been referred to as ‘nature’s adjuvants,’ and their use as cellular adjuvants has been explored to elicit potent protective immunity against both pathogens and tumors, with an antigen-presenting capacity that far surpasses that of standard adjuvant formulations [33].

In parallel, SsCWP has shown strong antigenic and protective properties in experimental models of *S. schenckii* infection [5], further supporting their use as promising candidates for DC-based vaccination strategies. Within this context, the use of SsCWP-stimulated DCs represents an attractive strategy to explore the balance between effector and regulatory immune responses in fungal vaccination models. In the present study, we observed that SsCWP extract promoted BMDC maturation by upregulating the expression of MHC II, CD83, CD80, CD86, and CD40 costimulatory molecules, as well as the production of pro-inflammatory cytokines such as TNF-α, IL-6, IL-12, and IL-1β. These molecules are essential for T cell proliferation and differentiation [34,35] and have been associated with protection against a wide range of mycoses, including sporotrichosis [15,16,36,37,38].

Before assessing the effect of BMDC vaccination in infected animals, we first evaluated the immunostimulatory capacity of BMDCs, either stimulated or not with SsCWP, in healthy BALB/c mice. By targeting highly immunogenic structural proteins [5], this strategy seeks to promote broader activation of lymphocyte populations while minimizing safety concerns, including the potential reversion of whole organisms to a viable fungal form. For this purpose, total splenocytes from uninfected animals immunized with SsCWP-stimulated or unstimulated BMDCs were analyzed for Th1 and Th17 cell frequencies and cytokine production. Immunization with SsCWP-stimulated BMDCs promoted a Th17-skewed immune response, evidenced by an increased frequency of IL-17A+ cells and elevated secretion of IL-17 and IL-6. Similar findings were reported by Verdan et al. [18], who showed that splenocytes from mice infected with *S. schenckii* and cocultured with BMDCs previously stimulated with either live yeast or fungal exoantigens released IL-17 and IFN-γ, consistent with a mixed Th1/Th17 profile. Th17 cells are known to be potent inducers of tissue inflammation and have been implicated in the pathogenesis of several autoimmune and inflammatory disorders. However, their primary function appears to be the elimination of pathogens that cannot be effectively controlled by Th1 or Th2 responses, such as fungal infections [36]. Indeed, we previously demonstrated that Th17-mediated immunity is essential for the effective clearance of *S. schenckii* in mice [15,17].

A significant increase in IL-10 was observed in the stimulated groups, concomitant with the enhanced production of pro-inflammatory cytokines (IL-17A, IFN-γ, IL-6). This pattern is consistent with an intrinsic compensatory mechanism, well documented in the literature, that protects the host from excessive inflammation [39,40]. This finding supports the potential of the vaccine candidate, which promotes a balanced response as indicated by the concurrent upregulation of both pro-inflammatory and regulatory cytokines.

Interestingly, a single immunization elicited stronger Th1/Th17 responses than two doses, which may reflect activation-induced tolerance or functional exhaustion. Repeated DC administration might engage regulatory pathways and negative feedback mechanisms, particularly if antigen presentation occurs without robust co-stimulatory signals or under the influence of immunoregulatory molecules such as IL-10 or TGF-β [41,42]. In this context, a single dose could represent a more favorable balance, sufficient to initiate protective immunity while reducing the likelihood of counter-regulatory responses. Nonetheless, this remains a tentative interpretation, and future studies will be needed to determine whether modifying booster intervals or dose spacing could further optimize the efficacy of DC-based immunotherapy.

After subcutaneous infection with *S. schenckii* in the footpad, mice exhibited a higher fungal burden in the draining popliteal lymph node than in the spleen, as expected given the proximity of the lymph node to the inoculation site. Because the inoculation was localized to the footpad, *S. schenckii* primarily entered and disseminated through the afferent lymphatics, seeding the draining (regional) lymph node first. Antigen-bearing DC and phagocytes can transport viable yeasts to this node, where early replication occurs before robust adaptive immunity develops. In contrast, the spleen is populated mainly via hematogenous spread; if bloodstream dissemination is limited or efficiently cleared by splenic macrophages, the splenic fungal burden remains lower at early time points. The draining lymph node is a preferential site for the induction of immune responses, and BMDCs administered subcutaneously are known to migrate there within 24 h [43,44,45], where they participate in the initiation of antifungal immunity. At the time of fungal burden evaluation, vaccination promoted a much more evident reduction in the spleen compared with the lymph node. The more pronounced reduction in fungal burden in the spleen suggests that vaccination induces systemic effects, highlighting the importance of DC migration and distribution in protective immunity. Future studies will assess long-term control of infection, with the expectation that sustained immunity will also achieve complete local control at the site of inoculation.

Our findings parallel those reported in the recent study by da Silva et al. [45], who demonstrated that extracellular vesicles released from DC, particularly when primed with *S. brasiliensis*, conferred significant protection against experimental sporotrichosis, reducing fungal load and promoting cytokine responses characterized by increased IFN-γ, TNF-α, IL-17, and IL-10. Similarly, in our study, vaccination with BMDCs, either unstimulated or stimulated with SsCWP, also led to a reduction in fungal burden and was associated with the induction of Th1- and Th17-skewed responses. Together, these observations reinforce the central role of dendritic cell–based immunomodulation in shaping protective immunity against *Sporothrix* spp. While da Silva et al. [45] explored a cell-free approach through DC-derived EVs, and we employed whole BMDCs, both strategies converge on enhancing Th1/Th17 responses as key correlates of protection.

Other comparable findings have been reported in studies employing DC-based vaccination strategies against a range of fungal pathogens. Notably, Ueno et al. [44] demonstrated that a DC-based vaccine, composed of BMDCs pulsed with a capsular *Cryptococcus gattii* antigens, induced long-lived lung-resident Th17 memory cells (TRM17). This approach not only reduced pulmonary fungal burden but also established durable protective immunity through tissue-resident memory. Similarly, our vaccination strategy using either unstimulated or SsCWP-stimulated BMDCs elicited robust Th1 and Th17 responses and effectively decreased fungal load in *S. schenckii*-infected mice. Additionally, Silva et al. reported that monocyte-derived dendritic cells (MoDCs), pulsed with the immunoprotective peptide P-10, promoted mixed Th1/Th2 cytokine responses and significantly reduced pulmonary fungal burden in a murine model of paracoccidioidomycosis [46]. This supports the notion that DC-based vaccination strategies can be broadly effective across diverse fungal pathogens, including those causing systemic mycoses.

In conclusion, our results highlight the pivotal role of DC in antifungal defense against *S. schenckii,* by acting as highly specialized antigen-presenting cells, inducing protective Th1/Th17 responses that are essential for fungal clearance. Our findings with SsCWP-stimulated BMDCs in *S. schenckii* infection, alongside recent studies employing DC-derived extracellular vesicles or DCs pulsed with fungal antigens in models of *Cryptococcus* and *Paracoccidioides* infections, underscore the broad applicability of DC-based immunotherapies across systemic mycoses. These observations not only validate DCs as central mediators of protective immunity but also reinforce their potential as therapeutic platforms for the development of next-generation antifungal vaccines.

## 5. Limitations and Future Directions

A limitation of our study is that the protective effect of BMDC-based vaccination was evaluated in a murine model under controlled experimental conditions, which may not fully recapitulate the complexity of human sporotrichosis. Moreover, while we focused on Th1 and Th17 responses, other immune mechanisms, including cytotoxic T lymphocytes, B cell responses, and regulatory pathways, were not explored in detail. Another limitation is the absence of long-term follow-up to determine whether protection is sustained beyond the acute phase of infection. Future studies should address these gaps by investigating memory responses, assessing vaccine efficacy in chronic or disseminated forms of sporotrichosis, and testing combination strategies with antifungal drugs. Exploring the translation of these findings to clinically relevant DC-based or cell-free platforms, such as dendritic cell-derived extracellular vesicles, may help to overcome the logistical challenges of cell-based vaccines and pave the way for novel immunotherapeutic approaches against *S. schenckii* and other pathogenic fungi. Such strategies could be particularly valuable in feline models, given the importance of cats as natural hosts and reservoirs of *Sporothrix* spp., and may ultimately support the development of DC-based or EV-based interventions for human sporotrichosis.

Despite feasibility constraints inherent to cell-based DC vaccines, our study provides valuable evidence for BMDC-based immunization against *S. schenckii*: both unstimulated and SsCWP-stimulated BMDCs elicited protective Th1/Th17 responses and reduced fungal burden, reinforcing the rationale for DC-centered immunotherapy in sporotrichosis and other systemic mycoses. To address translational barriers related to cost and logistics, we also highlight the potential of next-generation approaches, particularly cell-free modalities such as DC-derived extracellular vesicles and more scalable peptide-presenting platforms (e.g., peptide-loaded DCs or DC-mimetic systems), as promising strategies to overcome these limitations. 

## Figures and Tables

**Figure 1 microorganisms-13-02351-f001:**
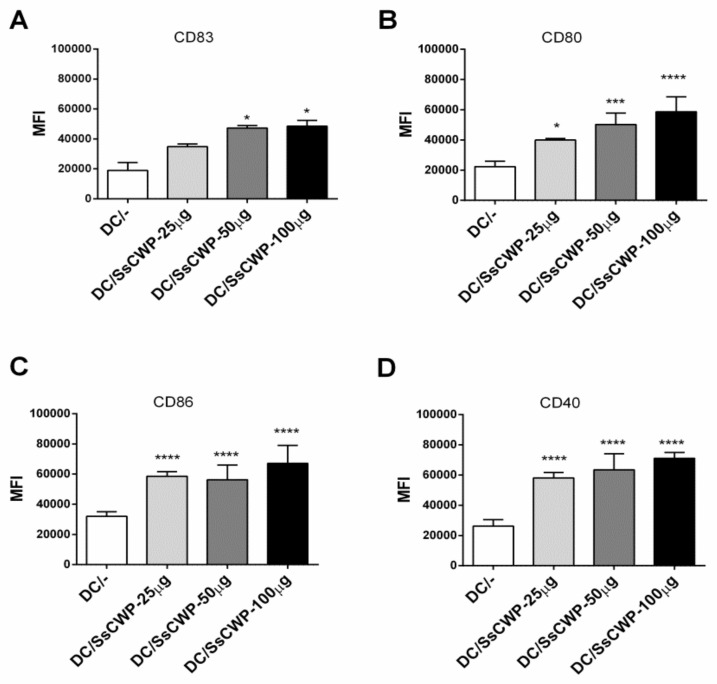
Expression of BMDC activation markers: (**A**) CD83, (**B**) CD80, (**C**) CD86, (**D**) CD40, before and after stimulation with SsCWP. The values correspond to three cultures, and the expression of cell activation markers is presented as the Median Fluorescence Intensity (MFI) of each marker. DC/-: Unstimulated BMDCs. DC/SsCWP: SsCWP-stimulated BMDCs. * (*p* < 0.05), *** (*p* < 0.001), **** (*p* < 0.0001): significantly higher when compared to unstimulated DC as indicated.

**Figure 2 microorganisms-13-02351-f002:**
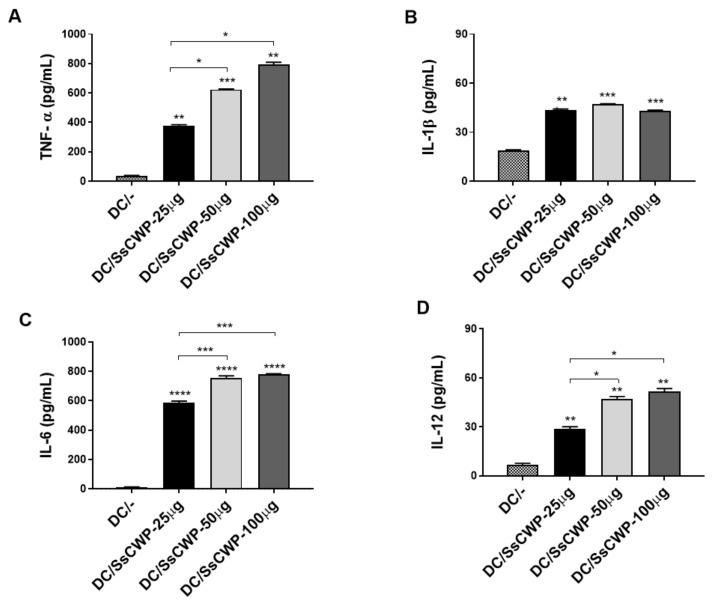
Quantification of cytokines in the culture supernatant of BMDCs stimulated or not with different concentrations of SsCWP: (**A**) TNF-α, (**B**) IL-1β, (**C**) IL-6, (**D**) IL-12. The values correspond to three cultures, and the results are expressed as the mean ± SD. DC/-: Unstimulated BMDCs. DC/SsCWP: SsCWP-stimulated BMDCs. Control (PBS). * (*p* < 0.05), ** (*p* < 0.01), *** (*p* < 0.001), **** (*p* < 0.0001) as indicated.

**Figure 3 microorganisms-13-02351-f003:**
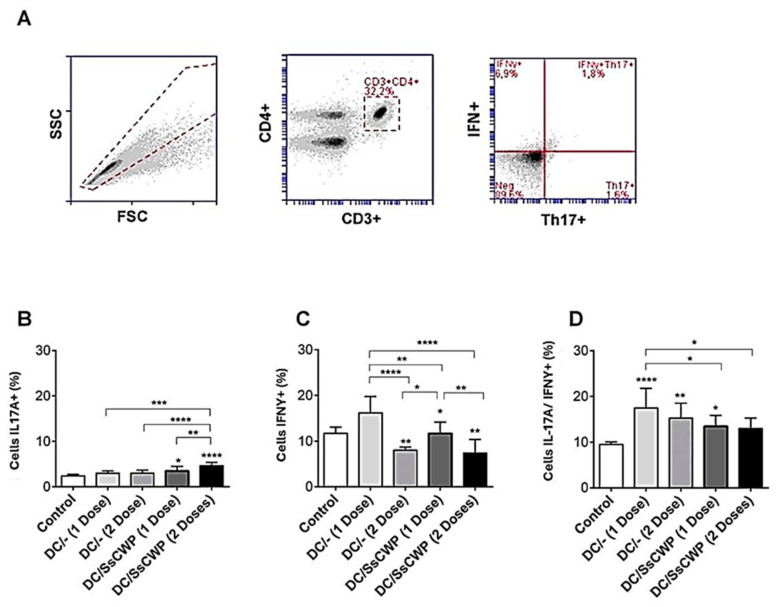
Frequency of Th17A + and IFN-γ + cell expression in splenocytes after 48 h culture in the presence of the heat-inactivated fungus *S. schenckii*: (**A**) Gating strategy, (**B**) expression IL-17A, (**C**) IFN-γ and (**D**) double population in mice vaccinated with 1 and 2 doses of SsCWP-stimulated or not stimulated BMDCs. Results are expressed as the mean ± SD of five animals. DC: Unstimulated BMDCs. DC/SsCWP: SsCWP-stimulated BMDCs. Control (PBS). * (*p* < 0.05), ** (*p* < 0.01), *** (*p* < 0.001), **** (*p* < 0.0001) as indicated.

**Figure 4 microorganisms-13-02351-f004:**
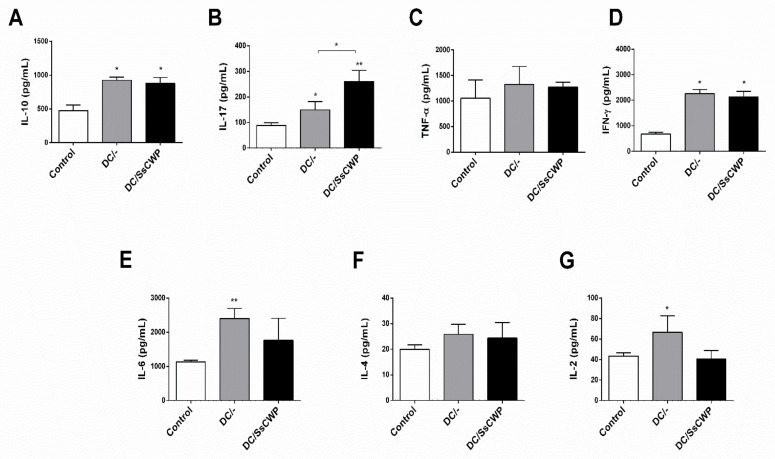
Quantification of cytokines in splenocyte culture supernatant after 48 h in the presence of the heat-inactivated fungus *S. schenckii*. (**A**) Cytokines IL-10, (**B**) IL-17, (**C**) TNF-α, (**D**) IFN-γ, (**E**) IL-6, (**F**) IL-4 and (**G**) IL-2 in vaccinated mice with BMDCs stimulated or not with SsCWP. Results are expressed as the mean ± SD of 5 animals. DC/-: Unstimulated BMDCs. DC/SsCWP: SsCWP-stimulated BMDCs. * (*p* < 0.05), ** (*p* < 0.01), significantly higher when compared to unstimulated DC or as indicated.

**Figure 5 microorganisms-13-02351-f005:**
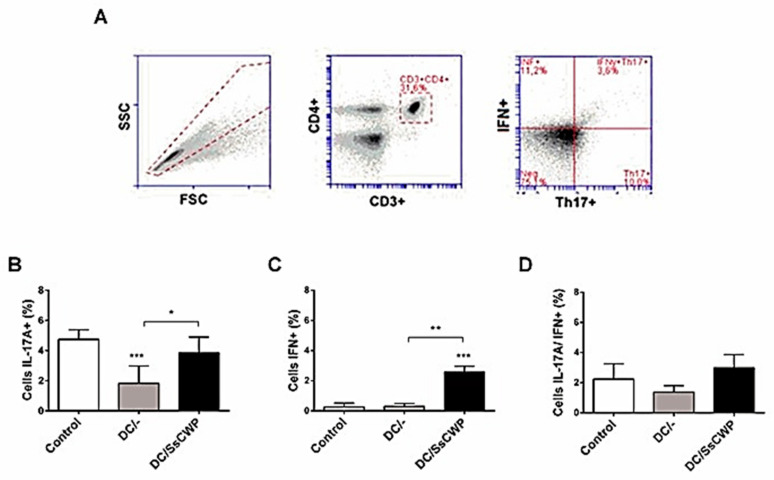
Frequency of Th17 + and IFN-γ + cell expression in the lymph node. (**A**) Gating strategy, % of (**B**) IL-17A, (**C**) IFN- γ, and (**D**) double positive population within the CD3^+^CD4^+^ splenocytes in *S. schenckii*-infected mice after vaccination with SsCWP-stimulated (DC/SsCWP), non-stimulated DCs (DC/-) or PBS (Control). Results are expressed as the mean ± SD of 5 animals. DC/-: Unstimulated BMDCs. DC/SsCWP: SsCWP-stimulated BMDCs, and Control (PBS) * (*p* < 0.05), ** (*p* < 0.01), *** (*p* < 0.001) as indicated.

**Figure 6 microorganisms-13-02351-f006:**
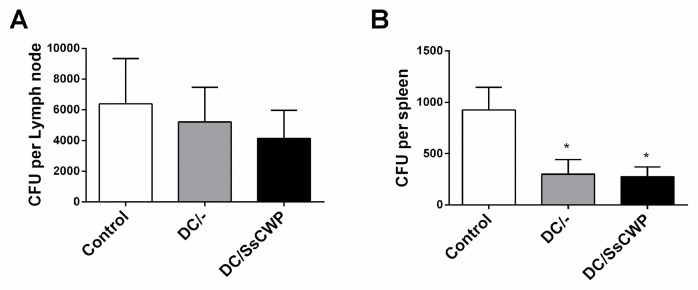
CFU recovered from the popliteal lymph node and spleen of *S. schenckii*-infected mice after vaccination with SsCWP-stimulated or non-stimulated DCs. (**A**) CFU per lymph node and (**B**) CFU per spleen. Results are expressed as the mean ± SD of 5 animals. DC/-: Unstimulated BMDCs. DC/SsCWP: SsCWP-stimulated BMDCs * (*p* < 0.05) were significantly lower when compared to the Control (PBS) group, as indicated.

## Data Availability

The original contributions presented in this study are included in the article/Appendix A. The data supporting the findings of this study are available in the original doctoral thesis by the author, which can be accessed at the following link: https://repositorio.unesp.br/server/api/core/bitstreams/856652ea-6d14-40c8-b49a-58a00560f091/content (accessed on 1 September 2025). Further inquiries can be directed to the corresponding authors.

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
