# Peer review of "Dendritic Cell-Based Therapeutic Immunization Induces Th1/Th17 Responses and Reduces Fungal Burden in Experimental Sporotrichosis"

_microorganisms, 2025, doi:10.3390/microorganisms13102351_

Round 1
Reviewer 1 Report
Comments and Suggestions for Authors
Make corrections as in this file

Author Response
We sincerely thank Reviewer 1 for the thoughtful comments and constructive suggestions, which have helped us to improve the clarity and depth of our manuscript.
All suggestions have been addressed in the main text, with changes highlighted in red.
Reviewer 2 Report
Comments and Suggestions for Authors
The manuscript entitled Dendritic Cell–Based Therapeutic Immunization Induces Th1/Th17 Responses and Reduces Fungal Burden in Experimental Sporotrichosis from Jellmayer et al. is a very compelling manuscript about DC based vaccine strategy for S. Schenckii. The group shows nicely that the model they have established to use BMDCs stimulated with SsCWPs effectively activate DCs and evaluation in an animal model shows lower burden in a two week experimental course of infection. Although the paper has great findings, some additional items are needed to really polish the study.
Some items that need improvement are:
- The paragraph from 3.1 and the 1st paragraph of 3.2 are identical and one should be removed.
- For the remainder of the results section, adding which figures the results are talking about rather than just the sub-figure letters will help the comprehension of the data.
- White boxes in figure 3 cut off part of the data.
- Having the clones of the antibodies in section 2.10 as was done in 2.7 would be of interest.
- Given the tough nature of intracellular cytokine staining, having the isotope controls in the flow gating (Figure 3A and Figure 4A) would help aid in seeing the positive cells
- It should be stated why the difference in the spleen undergoing 48 hours of culture (Figure 3 and Figure 5) where as the lymph node (Figure 4) it does not clarify how these cells were processed and if they underwent the same stimulation and Golgi blocking the splenocytes did.
Author Response
We sincerely thank Reviewer 2 for the careful evaluation of our work and for the valuable insights that guided meaningful revisions to the manuscript. All of the reviewers’ suggestions have been addressed in the main text, with the corresponding modifications clearly highlighted in red.
Reviewer: - The paragraph from 3.1 and the 1st paragraph of 3.2 are identical and one should be removed.
Response: Thank you for pointing this out. The duplicated paragraph has already been removed.
Reviewer: - For the remainder of the results section, adding which figures the results are talking about rather than just the sub-figure letters will help the comprehension of the data.
Response: We appreciate this suggestion. We have revised the Results section to indicate the corresponding figures explicitly, which should improve clarity and comprehension.
Reviewer:- White boxes in figure 3 cut off part of the data.
Response: Thank you for bringing this to our attention. The figure has been corrected and the white boxes have been removed to ensure all data are clearly visible.
Reviewer:- Having the clones of the antibodies in section 2.10 as was done in 2.7 would be of interest.
Response: We agree. The clones of the antibodies used in section 2.10 have now been added for consistency with section 2.7.
Reviewer: - Given the tough nature of intracellular cytokine staining, having the isotope controls in the flow gating (Figure 3A and Figure 4A) would help aid in seeing the positive cells
Response: We acknowledge the Reviewer’s point. We have now included the isotype controls in the gating strategy for Figures 3A and 4A to better illustrate the identification of positive cells.
Reviewer: - It should be stated why the difference in the spleen undergoing 48 hours of culture (Figure 3 and Figure 5) where as the lymph node (Figure 4) it does not clarify how these cells were processed and if they underwent the same stimulation and Golgi blocking the splenocytes did.
Response: We appreciate this comment. We have clarified in the Methods section that splenocytes used for cytokine release assays in supernatants were cultured independently, without the addition of monensin or brefeldin A. This distinction is now explicitly stated in the revised text
Reviewer 3 Report
Comments and Suggestions for Authors
-
Novelty and Significance
-
The study addresses an important gap in antifungal vaccine development, especially for Sporothrix schenckii.
-
However, the incremental novelty compared to previous work on dendritic cell (DC)–based fungal vaccines should be clarified. The authors should better highlight how their approach (SsCWP-stimulated BMDCs) differs from and advances beyond existing studies .
-
-
Experimental Design
-
The authors compared unstimulated vs. SsCWP-stimulated BMDCs, but the rationale for dose selection (50 μg/mL) should be justified more clearly. Was this based on dose–response optimization or prior publications?
-
The decision to use one vs. two immunizations should be discussed in more depth, as results suggest a single dose may be superior. What immunological mechanisms could explain reduced responses after a booster?
-
-
Immune Response Characterization
-
The study focuses heavily on Th1/Th17 cytokines, but lacks data on cytotoxic T cells, B cell/antibody responses, or regulatory T cells, which may influence fungal control. Expanding the immunophenotyping would strengthen conclusions.
-
The observed increase in IL-10 in vaccinated groups (Figure 5) is intriguing, but not sufficiently discussed. Could this reflect compensatory regulation that limits excessive inflammation?
-
-
Fungal Burden Analysis
-
Reduction in splenic fungal burden is convincing, but no significant change in lymph nodes raises questions. The authors should explain why systemic control is stronger than local control.
-
Were fungal burdens assessed beyond day 14? Without long-term data, durability of protection remains unclear .
-
-
Limitations and Translation
-
The study is limited to BALB/c mice and one fungal strain (ATCC 16345). Including more virulent isolates (e.g., S. brasiliensis) would strengthen the translational impact.
-
The authors should discuss feasibility and challenges of DC-based vaccines in clinical settings (e.g., cost, logistics, safety compared to subunit or EV-based vaccines).
-
Minor Comments
-
Clarity of Figures
-
Figures 1–6 contain valuable data, but some panels are overcrowded. Consider simplifying or moving detailed gating strategies to Supplementary Materials.
-
Statistical annotations should indicate exact n values per group in figure legends.
-
-
Writing and Flow
-
The Introduction is comprehensive but could be shortened by removing redundant explanations of DC biology .
-
The Discussion sometimes repeats results. Streamlining would improve readability.
-
-
References
-
References are generally up-to-date, but additional recent reviews on fungal immunotherapy and DC-based vaccines (2022–2024) could be cited. For example, Lionakis et al. (Nat Rev Immunol 2023) .
-
Comments on the Quality of English Language
C
omments on English Language and Style
-
General Clarity
-
The manuscript is understandable, but sentences are often long and contain multiple ideas. Shorter, more direct sentences would improve readability.
-
Some sections (e.g., the Introduction and Discussion) repeat similar information. Consider condensing to avoid redundancy.
-
-
Grammar and Syntax
-
Articles (“a,” “an,” “the”) are sometimes missing or used incorrectly. Example: “BMDC were cultured…” → should be “BMDCs were cultured…”.
-
Subject–verb agreement should be checked carefully. Example: “Data shows” → should be “Data show.”
-
Past tense should be used consistently for describing methods and results.
-
-
Word Choice
-
Phrases such as “make evident” or “showed expressive increase” could be replaced with more standard scientific terms: “demonstrate” or “showed a marked increase.”
-
Instead of “fungus burden,” use “fungal burden.”
-
-
Figures and Legends
-
Some figure legends are too brief. Expanding them with full explanations would help readers understand without referring back to the text.
-
Statistical notation (p-values, n) should follow journal style guidelines consistently.
-
-
Flow and Structure
-
The Results section occasionally mixes interpretation with description. Separate factual observations from interpretation for clarity.
-
The Discussion should move from key findings → interpretation → limitations → future perspectives, in a more structured way.
-
Author Response
We appreciate Reviewer 3’s detailed and constructive feedback, which has been instrumental in strengthening the overall quality of the manuscript. All the suggested revisions were incorporated into the main text, and the corresponding changes are indicated in red.
Reviewer 3
- Novelty and Significance
The study addresses an important gap in antifungal vaccine development, especially for Sporothrix schenckii. However, the incremental novelty compared to previous work on dendritic cell (DC)–based fungal vaccines should be clarified. The authors should better highlight how their approach (SsCWP-stimulated BMDCs) differs from and advances beyond existing studies .
Response: We thank the reviewer for this important comment. Our study differs from previous DC-based fungal vaccine approaches in several aspects:
- Unlike earlier studies that employed whole yeasts, exoantigens, or recombinant proteins of schenckii to stimulate DCs, we focused on bone marrow–derived DCs activated with a purified cell wall protein extract (SsCWP). By using structural proteins that are strongly immunogenic, this strategy aims to elicit a broader activation of specific lymphocyte populations while reducing safety concerns, such as the possibility of reversion to a viable fungal form when whole organisms are used.
- Importantly, our study is the first to assess the therapeutic (post-infection) effect of SsCWP-stimulated BMDCs, whereas most previous reports evaluated prophylactic settings.
- We demonstrate that SsCWP-stimulated BMDCs reduce systemic fungal burden while inducing a mixed Th1/Th17 response, which has not been clearly reported before in the context of schenckii.
We updated the Introduction and Discussion to emphasize these points more clearly.
- Experimental Design
The authors compared unstimulated vs. SsCWP-stimulated BMDCs, but the rationale for dose selection (50 μg/mL) should be justified more clearly. Was this based on dose–response optimization or prior publications? The decision to use one vs. two immunizations should be discussed in more depth, as results suggest a single dose may be superior. What immunological mechanisms could explain reduced responses after a booster?
Response: We thank the reviewer for these questions.
- The selection of 50 µg/mL SsCWP was based on preliminary dose–response experiments (25, 50, 100 µg/mL) showing that 50 µg/mL induced robust DC activation and cytokine secretion, with no further significant advantage at 100 µg/mL (Figure 2). We clarified this rationale in the Methods section.
- Regarding the immunization schedule, we observed that a single dose often induced stronger Th1/Th17 responses compared to two doses. This may reflect a phenomenon of activation-induced tolerance or exhaustion, as repeated DC injections can sometimes favor regulatory circuits or induce negative feedback (e.g., IL-10 upregulation)
- Immune Response Characterization
Reviewer: Lack of data on CTLs, B cells, Tregs. Also, IL-10 increase needs more discussion.
Response:
- We agree with the reviewer that additional immune subsets would provide a more complete picture. Our current focus was on Th1/Th17 responses, given their established role in schenckii clearance. Treg responses were not evaluated at the cellular level; however, measuring IL-10 concentrations provided an indirect approach to assess the activity of the regulatory pathway
We acknowledge this as a limitation and will explicitly state in the Limitations section that future studies will explore cytotoxic T cells, B-cell antibody production, and Treg contributions.
- The increase in IL-10 is indeed intriguing. We will expand the Discussion to suggest that IL-10 upregulation may represent a compensatory regulatory mechanism, potentially balancing inflammation to prevent immunopathology, as reported in other fungal infections.
- Fungal Burden Analysis
Reviewer: Why is systemic control stronger than local? Was long-term burden evaluated?
Response:
We observed a stronger reduction in splenic fungal burden compared to lymph nodes. Local footpad infection leads to predominant lymphatic seeding of the draining node, explaining its higher fungal burden relative to the spleen, which is mainly infected via limited hematogenous spread. DC immunization improved systemic control during the evaluation period, and upcoming long-term studies aim to confirm durable, comprehensive control—including at the local site. It was clarified in the discussion.
Fungal burden was assessed at day 14 post-infection, which was chosen to evaluate early therapeutic effects. We agree that long-term follow-up is essential to determine the durability of protection, and we will note this as a limitation and propose it for future studies.
- Limitations and Translation
Reviewer: Only BALB/c mice and one fungal strain; feasibility of DC-based vaccines.
Response:
- We acknowledge that our study was performed in BALB/c mice with a reference schenckii strain (ATCC 16345). Future work should include more virulent species such as S. brasiliensis to increase translational relevance. We will add this to the Limitations section.
- Regarding feasibility, we agree that cell-based DC vaccines face challenges in terms of cost, logistics, and clinical implementation. Our study represents a proof of concept. We updated the Discussion to emphasize the potential of cell-free alternatives (e.g., DC-derived extracellular vesicles and peptide-presenting platforms) as promising translational strategies to overcome these limitations.
Minor Comments
- Figures – We moved gating strategies and some crowded panels to Supplementary Materials, and specify exact n per group in figure legends.
- Writing and Flow – We updated the Introduction (removing redundant DC biology) and shortened the Discussion by reducing repetition.
- References – We add recent reviews such as Lionakis et al., Nat Rev Immunol 2023 and others from 2022–2024 on fungal immunotherapy.
English Language and Style
- We revised the manuscript for clarity, shortening long sentences and avoiding redundancy.
- Grammar corrections (articles, subject–verb agreement, consistent past tense) were applied.
- Word choice: replace “fungus burden” → “fungal burden,” “make evident” → “demonstrate,” etc.
- Figure legends were expanded with complete descriptions and statistical notation standardized.
- The Results were edited to separate observations from interpretations, and the Discussion will follow a structured flow (key findings → interpretation → limitations → future perspectives).